# Distributed Cubature Information Filtering Method for State Estimation in Bearing-Only Sensor Network

**DOI:** 10.3390/e26030236

**Published:** 2024-03-07

**Authors:** Zhan Chen, Wenxing Fu, Ruitao Zhang, Yangwang Fang, Zhun Xiao

**Affiliations:** Unmanned System Research Institute, Northwestern Polytechnical University, Xi’an 710072, China; chenzhan@mail.nwpu.edu.cn (Z.C.); ruitaozhang@mail.nwpu.edu.cn (R.Z.); ywfang@nwpu.edu.cn (Y.F.); xiaozhun@mail.nwpu.edu.cn (Z.X.)

**Keywords:** bearing-only sensor network, state estimation, DCIF algorithm, cooperative consistency theory

## Abstract

The problem of state estimation based on bearing-only sensors is increasingly important while existing research on distributed filtering solutions is rather limited. Therefore, this paper proposed the novel distributed cubature information filtering (DCIF) method for addressing the state estimation challenge in bearing-only sensor networks. Firstly, the system model of the bearing-only sensor network was constructed, and the observability of the system was analyzed. The sensor nodes are paired to measure relative angle information. Subsequently, the coordinated consistency theory is employed to achieve a unified state estimation of the maneuvering target. The DCIF method enhances the observability of the system, addressing the issues of large accuracy errors and divergence in traditional nonlinear filtering algorithms. Building upon the theoretical proof of consistency convergence in DCIF, four simulation experiments were conducted for comparison. These experiments validate the effectiveness and superiority of the DCIF method in bearing-only sensor networks.

## 1. Introduction

With the development of target tracking and locating technology based on various sensors, the research on state estimation methods as one of the core technologies is becoming more and more important; it can be divided into active sensors and passive sensors. The passive state estimation method mainly refers to the target tracking and positioning through the perception of the target electromagnetic or infrared signal by the sensor, including the signal radiated by the target and the signal reflected from the environment. This measurement technique does not require the sensor to actively emit electromagnetic or infrared signals but only needs to passively receive signals from the environment [1]. Compared with the observation mode of the active sensor, the most significant difference is that the information obtained by the passive sensor is limited and incomplete, which cannot satisfy the accuracy of target positioning. The bearing-only sensor is one of the most typical and widely used passive sensors.

The target information obtained by a single bearing-only sensor is limited and incomplete, which cannot satisfy the accuracy of target tracking. Therefore, in the practical application of bearing-only sensors, according to different information fusion processing methods, the configurations of sensors can be further divided into centralized, distributed and hybrid [2]. The traditional bearing-only sensor networks are mostly centralized processing. This method fully utilizes the information of all sensors and it can obtain higher accuracy. However, this method imposes a large computational burden on the central processing unit, and the system has low fault tolerance, resulting in poor adaptability in complex environments. The hybrid type also has the disadvantages of a centralized structure. Nowadays, distributed system structures with stronger advantages have been widely researched and applied. This configuration firstly processes the data of the sensor network locally and then fuses these local estimations non-centrally to obtain the global state estimation. It disperses the computational burden of the central node, enables parallel processing of data, and enhances the reliability and fault tolerance of the system while ensuring filtering accuracy [3].

Bearing-only sensors acquire target information by passively receiving azimuth and elevation angles from the target. They offer a long working range, strong concealment, and high adaptability to the working environment [4]. The use of bearing-only sensors reduces electromagnetic radiation, efficiently utilizes existing signal sources, and minimizes spectrum and energy requirements [5]. Its non-emitting characteristics improve electromagnetic compatibility and help reduce electromagnetic interference to other systems. On this basis, the multi-sensor system can obtain more accurate target status information and stronger stability through data fusion.

Especially in the coordinated detection of stealth maneuvering targets with excellent penetration capabilities, the blinding effect of stealth aircraft on active sensors such as radars and lasers can enable them to have superior penetration and confrontation capabilities [6]. Bearing-only sensors relying on pure angle measurement are the main solution in the current anti-stealth technology field because they only need to use the infrared or electromagnetic signals radiated by the stealth aircraft itself for detection [7]. The development of stealthy maneuvering targets has disrupted the existing balance of military confrontations. The imminent advancement of anti-stealth technology is crucial. Relevant academic research and engineering applications have been highly valued by the relevant departments of various countries in the world, and they are all competing to develop this cooperative state estimation technology based on a bearing-only sensor network.

In terms of academic research, the state estimation problem of nonlinear systems is solved by a variety of filter designs, mainly the Kalman Filter (KF) [8], Extended Kalman Filter (EKF) [9,10], Unscented Kalman Filter (UKF) [11,12], and Cubature Kalman Filter (CKF) [13,14]. These traditional methods all use the idea of KF to represent the system state and uncertainty with state vectors and covariance matrices. They are suitable for linear systems or approximately linear systems, with inevitable performance limitations for highly nonlinear systems that may require more computing resources, especially in high-dimensional state spaces. To improve the shortcomings of traditional filters, the Cubature Information Filter (CIF) was proposed by Chandra [15]. The CIF uses the Fisher information matrix (FIM) and information vector to represent system status and uncertainty [16]. Its nonlinear state estimation is better and easier to handle because it does not require a linearization process but directly operates on the information matrix (IM). Since then, CIF has gained significant traction in addressing estimation challenges posed by nonlinear systems [17]. On this basis, with the development of sensor network technology and information fusion technology, centralized CIF (CCIF) [18] and distributed CIF (DCIF) [19] algorithms have been further studied.

In engineering applications, CIF with better effects is applied to locate targets [20]. CIF is also widely used in the location of passive sensor networks in the fields of aviation [21], spaceflight [22] and navigation [23]. A distributed algorithm leveraging IMM was proposed by [24] to enhance the reliability of target tracking. The challenge of distributed network target tracking is tackled in [25] by employing the square-root CIF.

Through investigation and summarization, it has been observed that the CIF method is less commonly applied in the field of maneuvering targets compared to the KF and its improved versions. Moreover, research on efficient and stable DCIF in sensor networks is limited, especially in bearing-only sensor networks. It is necessary to consider how to design a locally applicable CIF that is suitable for observation constraints and to distribute it in a global manner to achieve DCIF. This paper focuses on the bearing-only sensor network and the distributed collaborative state estimation method, which is still a novel problem that urgently needs to be studied. The objective of this paper is to introduce a distributed state estimation technique designed for bearing-only sensor networks. This method aims to effectively and reliably track and locate maneuvering targets in a stable and efficient manner. The main innovative contributions are as follows.

To construct the positioning model based on a bearing-only sensor network and analyze the observability of the system;Based on observability analysis, the local CIF for a pair of bearing-only sensors is proposed and verified through simulation experiments;By designing the consistency protocol of the global network, the local CIF is extended to the entire bearing-only sensor network. The consistency of system convergence is proven, and simulation experiments are conducted to verify the effectiveness and superiority of the DCIF.

The remainder is organized as follows: Section 2 constructs a system model for target tracking through the angle of arrival (AOA) via a bearing-only sensor network, and performs the systematic observability analysis. In Section 3, the local CIF based on bearing-only sensor node pairs is proposed, and then the CIF is extended to the entire bearing-only sensor network through the collaborative consistency theory, and the DCIF algorithm suitable for distributed system network is obtained and proved. Section 4 substantiates the effectiveness and superiority of the DCIF method via four simulation experiments. Finally, the research content and future work directions of this paper are summarized.

Notation: This paper employs bold letters to signify vectors or matrices. Define $\gamma_{i}\in \left(-\pi ,\pi \right]$. E(·) denotes expectation.

## 2. System Model

The system model comprises three components. The initial component encompasses the measurement model of the bearing-only sensor pair within the local network. The second component entails the co-location model of the global bearing-only sensor network. The last one is the motion model describing the maneuvering target state.

### 2.1. Local Measurement Model of Bearing-Only Sensor Pair

In the process of cooperative detection for the target by the bearing-only sensor network model. Each sensor is capable of independently measuring both the azimuth and elevation angles with respect to the target, as shown in Figure 1. The positions of the bearing-only sensor pairs are $S_{i}\left(x_{i}\left(k\right),y_{i}\left(k\right),z_{i}\left(k\right)\right)$ and $S_{j}\left(x_{j}\left(k\right),y_{j}\left(k\right),z_{j}\left(k\right)\right)$, which measure the same target *T*. The elevation angle and azimuth angle measured by $S_{i}$ are $\theta_{i}$ and $\varphi_{i}$. Similarly, the two angles measured by $S_{j}$ are $\theta_{j}$ and $\varphi_{j}$.

The target is $X\left(t\right)={\left[x_{t}\left(k\right),y_{t}\left(k\right),z_{t}\left(k\right),\dot{x_{t}}\left(k\right),\dot{y_{t}}\left(k\right),\dot{z_{t}}\left(k\right),\ddot{x_{t}}\left(k\right),\ddot{y_{t}}\left(k\right),\ddot{z_{t}}\left(k\right)\right]}^{T}$, where x(t), y(t), z(t) are the coordinates of the target along the three axes, respectively. $\dot{x}\left(t\right)$, $\dot{y}\left(t\right)$, and $\dot{z}\left(t\right)$ represent the velocity of the target along the three axes, respectively. $\ddot{x}\left(t\right)$, $\ddot{y}\left(t\right)$, $\ddot{z}\left(t\right)$ represent the accelerations of the target along the three axes, respectively.

In three-dimensional space, the vectors of the bearing-only sensor node pair and the target are $\vec{OS_{i}}={\left(x_{i},y_{i},z_{i}\right)}^{T}$, $\vec{OS_{j}}={\left(x_{j},y_{j},z_{j}\right)}^{T}$ and $\vec{OS_{T}}={\left(x_{t},y_{t},z_{t}\right)}^{T}$. By utilizing the AOA measurement method [26], it is possible to derive the spatial coordinates of the target.
(1)$$\left\{\begin{array}{c} x_{t}=x_{i}+\frac{\left(y_{j}-y_{i}\right)\cos\varphi_{j}\cos\varphi_{i}-\left(x_{j}-x_{i}\right)\sin\varphi_{j}\cos\varphi_{i}}{\sin(\varphi_{i}-\varphi_{j})}\\ y_{t}=y_{i}+\frac{\left(y_{j}-y_{i}\right)\cos\varphi_{j}\sin\varphi_{i}-\left(x_{j}-x_{i}\right)\sin\varphi_{j}\sin\varphi_{i}}{\sin(\varphi_{i}-\varphi_{j})}\\ z_{t}=z_{i}+\frac{\left(y_{j}-y_{i}\right)\cos\varphi_{j}\sin\theta_{i}-\left(x_{j}-x_{i}\right)\sin\varphi_{j}\sin\theta_{i}}{\cos\theta_{i}\sin\left(\varphi_{i}-\varphi_{j}\right)} \end{array}\right.$$
when $\theta_{i}=\theta_{j}$ and $\varphi_{i}=\varphi_{j}$, the target line of sight of the sensor node pair coincides, the system of equations has no solution.

**Assumption** **1.**
*The triangular geometric relationship between the bearing-only sensor node pair and the target satisfies the requirement of state estimation.*


In the observation model of the bearing-only sensor network, two sensors and the target form a triangular geometric relationship for state estimation, as shown in Figure 1. In the formed triangle,
(2)$$\left\{\begin{array}{cc} R_{i} & =\frac{\sin\gamma_{j}}{\sin\gamma_{ij}}R_{ij}\\ R_{j} & =\frac{\sin\gamma_{i}}{\sin\gamma_{ij}}R_{ij}\\ \gamma_{ij} & =\pi -(\gamma_{i}+\gamma_{j}) \end{array}\right.$$
where $R_{i}$ represents the distance between $S_{i}$ and T. $R_{j}$ represents the distance between $S_{j}$ and T. $R_{ij}$ represents the distance between $S_{i}$ and $S_{j}$. $\gamma_{ij}$ represents the angle between the two bearing-only sensors and the target line of sight, which is called the line of sight separation (LOSS) angle. $\gamma_{i}$ is the angle between $S_{i}$ and T. $\gamma_{j}$ is the angle between $S_{j}$ and T.

The partial derivative of the $R_{i}$ for the *i*th bearing-only sensor node.
(3)$$\begin{array}{cc} \delta R_{i} & =\delta \left(\frac{\sin\gamma_{j}}{\sin\gamma_{ij}}R_{ij}\right)=\frac{\sin\gamma_{j}}{\sin\gamma_{ij}}\delta R_{ij}+\frac{R_{ij}\cos\gamma_{j}}{\sin\gamma_{ij}}\delta \gamma_{j}-\frac{R_{ij}\cos\gamma_{ij}\sin\gamma_{j}}{\sin^{2}\gamma_{ij}}\delta \gamma_{ij}\\ & =\frac{1}{\sin\gamma_{ij}}\left\{\sin\gamma_{j}\delta R_{ij}-R_{ij}\sin\gamma_{j}\cot\left(\gamma_{i}+\gamma_{j}\right)\delta \gamma_{i}+R_{ij}\left[\cos\gamma_{j}-\sin\gamma_{j}\cot\left(\gamma_{i}+\gamma_{j}\right)\right]\delta \gamma_{j}\right\} \end{array}$$

According to the analysis results, the estimation error of the *i*th bearing-only sensor for the distance is related to $\delta R_{ij}$, $\delta \gamma_{i}$ and $\delta \gamma_{ij}$. When the LOSS angle $\gamma_{ij}=\pi /2$, the distance estimation error is the smallest. When the LOSS angle reaches 0 or π, the distance estimation error reaches *∞*. At this time, the bearing-only sensors are in a straight line, the observation model degenerates into a single-sensor detection problem, the target observability is reduced, and the distance estimation cannot be achieved.

Therefore, in order to ensure that the bearing-only sensor can complete the estimation of the state information of the target, the triangular geometric relationship between the bearing-only sensor pair and the target must be satisfied. When the LOSS angle is closer to $\gamma_{ij}=\pi /2$, the distance estimation effect is better.

### 2.2. Distributed Bearing-Only Sensor Network Model

On the basis of analyzing the observation constraints of bearing-only sensors. The structure of the bearing-only sensor network is represented by an undirected connected graph G(N,R). R denotes the set of connections between bearing-only sensors. N is the set of bearing-only sensors. An edge (i,j)∈R indicates that $S_{j}$ can receive information from $S_{i}$. Further, for each bearing-only sensor i∈N, if sensor *j* is included in its neighbors, $N_{i}=\left\{j\left|(j,i)\in R\right.\right\}$. In other cases, $N_{i}\backslash \left\{j\right\}$ [27].

**Assumption** **2.**
*It is assumed that in the bearing-only sensor network model, each bearing-only sensor contains at least one neighbor sensor, and these two bearing-only sensors constitute a pair of the observation model. Therefore, the number of bearing-only sensors in the network should satisfy $N=\left\{1,2,\cdot \cdot \cdot ,N(N\geq 2)\right\}$.*


$S_{n}\left(x_{n}\left(k\right),y_{n}\left(k\right),z_{n}\left(k\right)\right),\left(n\in N,N\geq 2\right)$ are the positions of bearing-only sensors. They measure the same target source, and the estimated information of the target can be shared among adjacent bearing-only sensors; two bearing-only sensors constitute a pair [28]. According to the communication conditions within the bearing-only sensor network, $N_{n}$ bearing-only sensors can form *m* groups of bearing-only sensor pairs. Each pair of filtered results requires the help of consistency theory to ensure converging to the same estimation value. M is the set of bearing-only sensor pairs that satisfy the measurement requirements, it is expressed as
(4)$$i_{1},i_{2},\cdot \cdot \cdot ,i_{m}\left(m\leq \frac{N!}{2(N-2)!}\right)\in M\;\;\;\;N\geq 2$$
Each pair of bearing-only sensors will obtain local measurement information. The state observation of the system is $Z\left(k\right)={\left[\theta_{i}\left(k\right),\varphi_{i}\left(k\right),\theta_{j}\left(k\right),\varphi_{j}\left(k\right)\right]}^{T}$, where $\theta_{i}$, $\theta_{j}$ and $\varphi_{i}$, $\varphi_{j}$ represent the elevation and azimuth angles of the target measured by the two sensor nodes, respectively.

The observation equation of the system is
(5)$$Z\left(k\right)=h\left(X_{k}\right)+v\left(k\right)$$
According to the spatial position relationship in Figure 1, $h\left(X_{k}\right)$ can be obtained
(6) hθi(k)=arctanyt(k)−ySi(k)xt(k)−xSi(k) hφi(k)=arctanzt(k)−zSi(k)xt(k)−xSi(k)2+yt(k)−ySi(k)2 hθj(k)=arctanyt(k)−ySj(k)xt(k)−xSj(k) hφj(k)=arctanzt(k)−zSj(k)xt(k)−xSj(k)2+yt(k)−ySj(k)2
(7)$$v_{k}^{i}={\left[v_{\theta }^{i}\left(k\right),v_{\varphi }^{i}\left(k\right),v_{\theta }^{j}\left(k\right),v_{\varphi }^{j}\left(k\right)\right]}^{T}$$
where, $v_{\theta }^{i}\left(k\right)$ and $v_{\varphi }^{i}\left(k\right)$ represent the measurement errors of $S_{i}$ in the the elevation and azimuth, respectively. $v_{\theta }^{j}\left(k\right)$ and $v_{\varphi }^{j}\left(k\right)$ represent the measurement errors of $S_{j}$ in the elevation and azimuth, respectively.

How to fuse this local information into a unified global information in a bearing-only sensor network is a distributed information fusion problem. This paper completes this part of the work by designing a weighted average consistency protocol.

The term ${\left(Y_{k|k}^{i},L_{k|k}^{i}\right)}_{i\in I}$ is defined as an information pair. When *ℓ* approaches *∞* and each information pair tends towards a uniform value, it is considered to have achieved weighted average consensus.
(8)$$\left({\hat{Y}}_{k}^{*},L_{k}^{*}\right)=\lim_{\ell \to \infty }\;\left({\hat{Y}}_{k,\ell }^{i},L_{k,\ell }^{i}\right)$$
where, ${\left({\hat{Y}}_{k}^{i},L_{k}^{i}\right)}_{i\in I}$ represent information pairs accessible at node *i* during the *l*th iteration and meet the condition: (9)$$\left\{\begin{array}{c} {\hat{Y}}_{k,\ell +1}^{i}=\sum_{j\in I_{i}}\pi^{ij}{\hat{Y}}_{k,\ell }^{j}\\ L_{k,\ell +1}^{i}=\sum_{j\in I_{i}}\pi^{ij}L_{k,\ell }^{j} \end{array}\right.$$
where $\pi^{ij}\geq 0$, $\sum_{j\in I_{i}}\pi^{ij}=1$ and the initial values are ${\hat{Y}}_{k,0}^{i}={\hat{Y}}_{k}^{i}$, $L_{k,0}^{i}=L_{k}^{i}$, then the information pairs reach the weighted average consistency. The bearing-only sensor network can get a uniform estimation result.

The type of sensors under investigation in this paper is passive bearing-only sensors, which have broad applicability. Passive radars and infrared sensors are examples of bearing-only sensors. In this study, we have chosen to focus on bearing-only sensors, rather than specific types for two main reasons. Firstly, the constructed system model is highly correlated with the angular measurement mechanism of the sensors. If different types of bearing-only sensors are used, only different angular measurement errors need to be set, which will not affect the research results of the system. Secondly, describing the research object as bearing-only sensors can provide technical references for a wider range of related applications, making the model and distributed methods constructed in this paper more universally applicable. It is important to highlight that this universality assumption holds true under the condition of a homogeneous bearing-only sensor network.

### 2.3. Motion Model of the Maneuvering Target

Establishing a reasonable dynamic model of maneuvering targets significantly influences the accuracy of state estimation. The maneuvering target addressed in this paper primarily refers to aircraft utilizing solid fuel as their propellant. These aircraft have simple structures, eliminating the need for complex fuel supply systems and liquid fuel handling equipment, making them easy to load. However, the combustion process of such maneuverable targets is challenging to control, making mid-course thrust adjustments difficult [29]. Therefore, the use of the Singer motion model provides a vivid description of their dynamics.

The Singer model of the maneuvering target is a stationary random process, which can reliably describe the maneuver characteristics of the target. The model represents the target acceleration as a zero-mean random process with exponential autocorrelation, and the exponential decay form of the time correlation function is expressed as
(10)$$R_{\mu }\left(\delta \right)=E\left[a\left(t\right)a\left(t+\delta \right)\right]=\sigma^{2}e^{-\mu |\delta |},\mu \geq 0$$
where σ and μ are undetermined parameters that determine the maneuvering characteristics of the target within $\left[t,t+\delta \right]$. $\sigma^{2}$ is the variance of the acceleration. μ represents the frequency, defined as the reciprocal of the maneuvering time constant. The probability distribution of acceleration is as follows.

There is a probability $P_{0}$ that the target moves without acceleration;There is a probability $P_{\max}$ that the target moves with the maximum acceleration $a_{\max}$, and also the probability $P_{\max}$ of moving with the minimum acceleration $-a_{\max}$;The target maneuvering acceleration approximately obeys a uniform distribution in the $\left[-a_{\max},a_{\max}\right]$.

Calculate the variance of the available acceleration as
(11)$$\sigma^{2}=\frac{a_{\max}^{2}}{3}\left(1+4P_{\max}-P_{0}\right)$$
where $P_{0},P_{\max},a_{\max}$ and μ are all parameters of the prior design [29].

Then the equation of motion in continuous time is
(12)$$\hat{X}\left(t\right)=AX\left(t\right)+B\omega \left(t\right)$$
where ω(t) is white Gaussian noise with zero mean.
(13)$$A=diag\left(\left[\begin{array}{ccc} 0 & 1 & 0\\ 0 & 0 & 1\\ 0 & 0 & -\mu \end{array}\right]\;,\;\left[\begin{array}{ccc} 0 & 1 & 0\\ 0 & 0 & 1\\ 0 & 0 & \mu \end{array}\right]\;,\;\left[\begin{array}{ccc} 0 & 1 & 0\\ 0 & 0 & 1\\ 0 & 0 & \mu \end{array}\right]\right)$$
(14)$$B=diag\left({\left[\begin{array}{ccc} 0 & 0 & 1 \end{array}\right]}^{T},{\left[\begin{array}{ccc} 0 & 0 & 1 \end{array}\right]}^{T},{\left[\begin{array}{ccc} 0 & 0 & 1 \end{array}\right]}^{T}\right)$$

After the continuous time system is discretized with the sampling period *T*, according to the optimal filtering theory, it is transformed into the equation of state in discrete time.
(15)$$X_{k}=f\left(X_{k-1}\right)+w_{k-1}=\Phi_{k\left|k-1\right.}X_{k-1}+w_{k-1}$$
where $\Phi_{k\left|k-1\right.}$ is the state transition matrix.
(16)$$\Phi_{k\left|k-1\right.}=diag\left(F,F,F\right)$$
(17)$$F=\left[\begin{array}{ccc} 1 & T & \left(\mu T-1+e^{-\alpha T}\right)/\mu^{2}\\ 0 & 1 & \left(1-e^{\mu T}\right)/\mu \\ 0 & 0 & -e^{\mu T} \end{array}\right]$$

The variance is $Q_{k-1}$.
(18)$$Q_{k-1}=diag\left(Q,Q,Q\right)$$

*Q* is a symmetric matrix, where
(19)$$Q=2\mu \sigma^{2}\left[\begin{array}{ccc} q_{11} & q_{12} & q_{13}\\ q_{21} & q_{22} & q_{23}\\ q_{31} & q_{32} & q_{33} \end{array}\right]$$
where,
(20)$$\left\{\begin{array}{c} q_{11}=\left(2\mu^{3}T^{3}-6\mu^{2}T^{2}+6\mu T+3-12\mu Te^{-\mu T}-3e^{-2\mu T}\right)/\left(6\mu^{5}\right)\\ q_{12}=q_{21}=\left(\mu^{2}T^{2}-2\mu T+1-2\left(1-\mu T\right)e^{-\mu T}+e^{-2\mu T}\right)/\left(2\mu^{4}\right)\\ q_{22}=\left(2\mu T-3+4e^{-\mu T}-e^{-2\mu T}\right)/\left(2\mu^{3}\right)\\ q_{13}=q_{31}=\left(1-2\mu Te^{-\mu T}-e^{-2\mu T}\right)/\left(2\mu^{3}\right)\\ q_{23}=q_{32}=\left(1-2e^{-\mu T}+e^{-2\mu T}\right)/\left(2\mu^{2}\right)\\ q_{33}=\left(1-e^{-2\mu T}\right)/\left(2\mu \right) \end{array}\right.$$

Combining the above models, the system model can be obtained.
(21)$$\left\{\begin{array}{c} X_{k}=f\left(X_{k-1}\right)+w_{k-1}\\ Z_{k}^{i}=h^{i}\left(X_{k}\right)+v_{k}^{i},\; \end{array}\right.\;\;i\in M_{i}$$

**Assumption** **3.**
*T is the sampling period, it is assumed that the measurement period of all the bearing-only sensors is the same, and the measured data are aligned in space and time.*


## 3. The Proposed Algorithms

The proposed DCIF method is used to perform filtering iterations at each node pair. Only the locally measured information of the bearing-only sensor pairs is needed, and the final convergent and unified information with the help of the cooperative consistency theory can be obtained. The proposed DCIF method offers several advantages in terms of robustness, accuracy, scalability, and adaptability. By leveraging volumetric information and facilitating collaboration among networked sensors, this method addresses some of the limitations of traditional approaches and provides a more comprehensive framework for target tracking in a bearing-only sensor network.

### 3.1. CIF Based on Bearing-Only Sensor Pair

The bearing-only sensor network and the Singer-type maneuvering target studied in this paper are often applied in denial environments with complex interference conditions. Considering constraints on real-time performance and computational capability, the CIF is well-suited to address this problem. However, how to design the CIF suitable for the system model in this study is essential to ensure that the measurement model of CIF satisfies the constraints of pairwise observations.

CIF is a state estimation algorithm for nonlinear systems. It combines the information matrix concept of Kalman filtering with the characteristics of Gaussian integration, aiming to better preserve the characteristics of the uncertainty distribution and thus handle state estimation problems in nonlinear systems more effectively [16].

CIF estimates and updates the state by manipulating the information matrix. The FIM reflects the uncertainty and precision associated with the state estimation problem. At the beginning of CIF, the next state is predicted based on the motion model. The prediction process involves the transformation of Cubature Points (CPs) and the calculation of weights [19]. When new measurement data are available, CIF calculates the value of CPs on the observation function, as well as the information covariance matrix between the estimated state and the measurement. Then, the measurement update of the state is performed by associating the transformation and weight of the CPs with the measurement.

For each pair of bearing-only sensor nodes *s*, the CIF algorithm comprises two stages: the update time stage and the update measurement stage.

#### 3.1.1. The Update Time Stage

Let m=2n CPs denoted as $X_{k-1|k-1}^{s,i}\in R^{n}$, $X_{k-1|k-1}^{s,i}$ is generated by estimated value ${\hat{X}}_{k-1|k-1}^{s}$ and the square root matrix $S_{k-1|k-1}^{s}$, where $S_{k-1|k-1}^{s}$ is the square of the matrix $L_{k-1|k-1}^{s}$.
(22)$$X_{k-1|k-1}^{s,i}=S_{k-1|k-1}^{s}\cdot \xi_{i}+{\hat{X}}_{k-1|k-1}^{s}\;\left(i\in M_{i}\right)$$
(23)$$\xi_{i}=\left\{\begin{array}{cc} \sqrt{n}e_{i}, & i\in \left[1,n\right]\\ -\sqrt{n}e_{i-n}, & i\in \left[n+1,m\right] \end{array}\right.$$

$e_{i}$ represents an *n*-dimensional unit vector where its *i*-th element is equal to 1.
(24)$$L_{k-1|k-1}^{s}=E\left[\left(X_{k-1}^{s}-{\hat{X}}_{k-1}^{s}\right){\left(X_{k-1}^{s}-{\hat{X}}_{k-1}^{s}\right)}^{T}\right]$$

Each CP $X_{k-1|k-1}^{s,i}$ is mapped to the following point through $p\left(\cdot \right)$.
(25)$$X_{k|k-1}^{*s,i}=p\left(X_{k-1|k-1}^{s,i}\right)\in R^{n}\;\left(i\in M_{i}\right)$$

Therefore, the predicted state ${\hat{X}}_{k|k-1}^{s}$, predicted IM $L_{k|k-1}^{s}$, and predicted information state ${\hat{Y}}_{k|k-1}^{s}$ are
(26)$$\left\{\begin{array}{c} {\hat{X}}_{k|k-1}^{s}=\frac{1}{m}\sum_{i=1}^{m}X_{k|k-1}^{*s,i}\\ L_{k|k-1}^{s}={\left[\frac{1}{m}\sum_{i=1}^{m}X_{k|k-1}^{*s,i}{\left(X_{k|k-1}^{*s,i}\right)}^{T}-{\hat{X}}_{k|k-1}^{s}{\left({\hat{X}}_{k|k-1}^{s}\right)}^{T}+\theta_{k-1}\right]}^{-1}\\ {\hat{Y}}_{k|k-1}^{s}=L_{k|k-1}^{s}{\hat{X}}_{k|k-1} \end{array}\right.$$

#### 3.1.2. The Update Measurement Stage

Firstly, generate a new set of CPs $X_{k|k-1}^{s,i}\in R^{n}$ based on the predicted state ${\hat{X}}_{k|k-1}^{s}$ and $S_{k-1|k-1}^{s}$, satisfies $S_{k-1|k-1}^{s}{\left(S_{k-1|k-1}^{s}\right)}^{T}={\left(L_{k|k-1}^{s}\right)}^{-1}$.
(27)$$X_{k|k-1}^{s,i}=S_{k|k-1}^{s}\xi_{i}+{\hat{X}}_{k|k-1}^{s}\;\left(i\in M_{i}\right)$$

Subsequently, propagate the CPs using the measurement function $h^{s}\left(X_{k}\right)$.
(28)$$Z_{k|k-1}^{s,i}=h^{s}\left(X_{k|k-1}^{s,i}\right)\;\left(i\in M_{i}\right)$$

Thus, the predicted values ${\hat{Z}}_{k|k-1}^{s,i}$ is:(29)$${\hat{Z}}_{k|k-1}^{s,i}=\frac{1}{m}\sum_{i=1}^{m}Z_{k|k-1}^{s,i}$$

The information state distribution $i_{k}^{s}$ and the corresponding IM $I_{k}^{s}$ are
(30)$$\left\{\begin{array}{c} i_{k}^{s}=L_{k|k-1}^{s}P_{xz,k|k-1}^{s}{\left(R_{k}^{s}\right)}^{-1}\left[V_{k}^{s}+{\left(P_{xz,k|k-1}^{s}\right)}^{T}L_{k|k-1}^{s}{\hat{X}}_{k|k-1}^{s}\right]\\ I_{k}^{s}=L_{k|k-1}^{s}P_{xz,k|k-1}^{s}{\left(R_{k}^{s}\right)}^{-1}{\left(P_{xz,k|k-1}^{s}\right)}^{-1}L_{k|k-1}^{s} \end{array}\right.$$
(31)$$P_{xz,k|k-1}^{s}=\frac{1}{m}\sum_{i=1}^{\infty }X_{k|k-1}^{s,i}{\left(Z_{k|k-1}^{s,i}\right)}^{T}-{\hat{X}}_{k|k-1}^{s}{\left({\hat{Z}}_{k|k-1}^{s}\right)}^{T}$$
(32)$$V_{k}^{s}=Z_{k}^{s}-{\hat{Z}}_{k|k-1}^{s}$$

Finally, ${\hat{Y}}_{k|k}^{s}$, $L_{k|k}^{s}$, and ${\hat{X}}_{k|k}^{s}$ can be obtained.
(33)$$\left\{\begin{array}{c} {\hat{Y}}_{k|k}^{s}=L_{k|k-1}^{s}+i_{k}^{s}\\ L_{k|k}^{s}=L_{k|k-1}^{s}+I_{k}^{s}\\ {\hat{X}}_{k|k}^{s}={\left(L_{k|k}\right)}^{-1}{\hat{Y}}_{k|k}^{s} \end{array}\right.$$

### 3.2. DCIF in Bearing-Only Sensor Network

Distributed filtering methods represent all bearing-only sensors as nodes, and information transfer among nodes is depicted by a topology graph. In the distributed network, each bearing-only sensor node shares state estimation with its neighbors, and the local estimates are unified. Eventually, all the local estimates converge to a consistent result. The proposed DCIF method is based on the cooperative consistency theory. The process of weighted average ensures the coherence of state estimations across all sensor nodes and finally obtains the unified result.

The communication conditions satisfied are that the information exchanges are only carried out among the adjacent nodes, and the local information is solely dependent on the received information. The flowchart of the DCIF method is shown in Figure 2.

The specific steps for DCIF to achieve consistent state estimation are as follows.

Calculate the initial information pairs $\left(Y_{k,0}^{s},L_{k,0}^{s}\right)$ of each node pair $s\in M_{s}$;
(34)$$\left\{\begin{array}{c} {\hat{Y}}_{k,0}^{s}={\hat{Y}}_{k|k-1}^{s}+L_{k|k-1}^{s}P_{xz,k|k-1}^{s}{\left(R_{k}^{s}\right)}^{-1}\left[V_{k}^{s}+{\left(P_{xz,k|k-1}^{s}\right)}^{T}L_{k|k-1}^{s}{\hat{X}}_{k|k-1}^{s}\right]\\ {\hat{L}}_{k,0}^{s}=L_{k|k-1}^{s}+L_{k|k-1}^{s}P_{xz,k|k-1}^{s}{\left(R_{k}^{s}\right)}^{-1}{\left(P_{xz,k|k-1}^{s}\right)}^{T}L_{k|k-1}^{s} \end{array}\right.$$For l=0,1,···,L−1, the following steps for running the protocol for weighted average consistency;Broadcast message $\left(Y_{k,l}^{s},L_{k,l}^{s}\right)$ to neighbor $j\in M_{s}$;Receive information $\left(Y_{k,l},L_{k,l}^{j}\right)$ from all neighbors $j\in M_{s}$;The received information is fused according to Equations (8) and (9).Update state information estimation and information matrix estimation;
(35)$$\left\{\begin{array}{c} {\hat{Y}}_{k|k}^{s}={\hat{Y}}_{k,L}^{s},\;L_{k|k}^{s}=L_{k,L}^{s}\\ {\hat{X}}_{k|k}^{s}={\left(L_{k|k}^{s}\right)}^{-1}{\hat{Y}}_{k|k}^{s} \end{array}\right.$$Finally, the information state vector estimate at step k+1 is calculated and obtained.
(36)$$\left\{\begin{array}{c} {\hat{X}}_{k+1|k}^{s}=\frac{1}{m}\sum_{i=1}^{m}X_{k+1|k}^{*s,i}\\ L_{k+1|k}^{s}={\left[\frac{1}{m}\sum_{i=1}^{m}X_{k+1|k}^{*s,i}{\left(X_{k+1|k}^{*s,i}\right)}^{T}-{\hat{X}}_{k+1|k}^{s}{\left({\hat{X}}_{k+1|k}^{s}\right)}^{T}+\theta_{k}\right]}^{-1}\\ {\hat{Y}}_{k+1|k}^{s}=L_{k+1|k}^{s}{\hat{X}}_{k+1|k}^{s} \end{array}\right.$$

The weighted average consistency protocol assigns weights to each observation based on factors such as the reliability of measurements from bearing-only sensors, the consistency of observations between pairs of sensors, and the estimated uncertainty associated with each observation. Observations deemed more reliable and consistent could receive higher weights, while those considered outliers are treated as such. Outliers are removed to prevent them from unduly influencing the estimation process. Each observation is multiplied by its respective weight, and the weighted observations are then summed to generate a fused estimate of the state. This fusion process ensures that observations are integrated in a consistent and robust manner, taking into account their respective uncertainties and levels of reliability.

The proposed DCIF is based on the weighted average consistency protocol, allowing the state estimation of the system to converge to a unified result within a controllable finite time.

**Assumption** **4.**
*During the convergence process of DCIF, if the initial prediction estimate ${\left\{{\hat{X}}_{1|0}^{s}\right\}}_{s=1}^{M}$ exhibits consistency, that is*

(37)
$$L_{1|0}^{s}=E\left[\left(X-{\hat{X}}_{1|0}^{s}\right){\left(X-{\hat{X}}_{1|0}^{s}\right)}^{T}\right]$$

*For each k>1, s∈M, $L_{k|k-1}^{s}\leq {\left\{E\left[\left(X_{k}-{\hat{X}}_{k|k-1}^{s}\right){\left(X_{k}-{\hat{X}}_{k|k-1}^{s}\right)}^{T}\right]\right\}}^{-1}$ and $L_{k|k}^{s}\leq {\left\{E\left[\left(X_{k}-{\hat{X}}_{k|k-1}^{s}\right){\left(X_{k}-{\hat{X}}_{k|k}^{s}\right)}^{T}\right]\right\}}^{-1}$, the DCIF method maintains consistency.*


**Proof of Assumption** **4.****Theorem** **1.**
*$\hat{X}$ is an estimate vector of X. P is the corresponding covariance of the error. If $E\left[\left(X-\hat{X}\right){\left(X-\hat{X}\right)}^{T}\right]\leq P$, then the pair $\left(\hat{X},P\right)$ exhibits consistency [24].*
Let the covariance between the prior estimate error of bearing-only sensor node pair *s* and the actual prior error be
(38)$$\left\{\begin{array}{c} {\hat{X}}_{k|k-1}^{s}≜X_{k}-{\hat{X}}_{k|k-1}^{s}\\ {\hat{P}}_{k|k-1}^{s}≜E\left[{\hat{X}}_{k|k-1}^{s}{\left({\hat{X}}_{k|k-1}^{s}\right)}^{T}\right] \end{array}\right.$$
Define the covariance between ${\hat{X}}_{k}^{s}$ and ${\hat{P}}_{k|k}^{s}$ as
(39)$$\left\{\begin{array}{c} {\hat{X}}_{k}^{s}≜X_{k}-{\hat{X}}_{k|k}^{s}\\ {\hat{P}}_{k|k}^{s}≜E\left[{\hat{X}}_{k}^{s}{\left({\hat{X}}_{k}^{s}\right)}^{T}\right] \end{array}\right.$$
The cross-covariance matrix can be approximated as
(40)$$P_{xz,k|k-1}^{s}=E\left[\left(X_{k}-X_{k|k-1}^{s}\right){\left(Z_{k}-{\hat{Z}}_{k|k-1}^{s}\right)}^{T}\right]\approx {\left(L_{k|k-1}^{s}\right)}^{-1}{\left(H_{k}^{s}\right)}^{T}$$
where, $H_{k}^{s}≜\partial h^{s}\left(X\right)/{\left.\partial X\right|}_{X={\hat{X}}_{k|k-1}^{s}}$. Then there is ${\tilde{H}}_{k}^{s}$ as
(41)$${\tilde{H}}_{k}^{s}={\left(P_{xz,k|k-1}^{s}\right)}^{T}L_{k|k-1}^{s}$$
Similarly, there is
(42)$$P_{X_{k-1},X_{k|k-1}}^{s}=E\left[\left(X_{k-1}-{\hat{X}}_{k-1|k-1}^{s}\right){\left(X_{k}-{\hat{X}}_{k|k-1}^{s}\right)}^{T}\right]=P_{k-1|k-1}^{s}{\left(F_{k-1}^{s}\right)}^{T}$$
where, $F_{k-1}^{s}≜\partial f\left(X\right)/{\left.\partial X\right|}_{X={\hat{X}}_{k-1|k-1}^{s}}$.Therefore, define the pseudosystem matrix as
(43)$${\tilde{F}}_{k-1}^{s}≜{\left(P_{k-1|k-1}^{s}\right)}^{T}L_{k-1|k-1}^{s}$$
where, $L_{k-1|k-1}^{s}={\left(P_{k-1|k-1}^{s}\right)}^{-1}$. $P_{X_{k-1},X_{k|k-1}}$ is
(44)$$P_{X_{k-1},X_{k|k-1}}^{s}=\frac{1}{m}\sum_{i=1}^{m}\left(X_{k-1|k-1}^{s,i}-{\hat{X}}_{k-1|k-1}^{s}\right){\left(X_{k|k-1}^{*s,i}-{\hat{X}}_{k|k-1}^{s}\right)}^{T}$$
Hence, the nonlinear bearing-only-only sensor network can be linearized into the following model within a discrete-time system.
(45)$$\left\{\begin{array}{c} X_{k}=\alpha_{k-1}^{s}{\tilde{F}}_{k-1}X_{k-1}+W_{k-1}\\ X_{k}^{s}=\beta_{k}^{s}{\bar{H}}_{k}^{s}X_{k}+V_{k}^{s} \end{array}\right.\;\left(s\in M\right)$$
where, the unknown auxiliary matrices $\alpha_{k}^{s}=\mathrm{diag}\left(\alpha_{k,1}^{s},\cdots ,\alpha_{k,n}^{s}\right)$, $\beta_{k}^{s}=\mathrm{diag}\left(\beta_{k,v}^{s},\cdots ,\beta_{k,r}^{s}\right)$ are used to compensate for approximation errors during the linearization process. Therefore, by rearranging the equations, the following equation could be obtained.
(46)$$\left\{\begin{array}{c} {\tilde{P}}_{k|k}^{s}=\left(I_{n}-W_{k}^{s}\beta_{k}^{s}{\tilde{H}}_{k}^{s}\right){\tilde{P}}_{k|k-1}^{s}{\left[\left(I_{n}-W_{k}^{s}{\mathrm{fi}}_{k}^{s}{\tilde{H}}_{k}^{s}\right){\tilde{P}}_{k|k-1}^{s}\right]}^{T}+W_{k}^{s}R_{k}^{s}{\left(W_{k}^{s}R_{K}^{s}\right)}^{T}\\ P_{k|k}^{s}=\left(I_{n}-W_{k}^{s}{\mathrm{fi}}_{k}^{s}{\tilde{H}}_{k}^{s}\right){\tilde{P}}_{k|k-1}^{s}{\left[\left(I_{n}-W_{k}^{s}{\mathrm{fi}}_{k}^{s}{\tilde{H}}_{k}^{s}\right){\tilde{P}}_{k|k-1}^{s}\right]}^{T}+W_{k}^{s}R_{k}^{s}{\left(W_{k}^{s}R_{k}^{s}\right)}^{T} \end{array}\right.$$
where $W_{k}^{s}$ is the gain. $P_{k|k}^{s}={\left(L_{k|k}^{s}\right)}^{-1}$.Assuming that at each time step *k*, there is
(47)$$L_{k|k-1}^{s}\leq {\left\{E\left[\left(X_{k}-{\hat{X}}_{k|k-1}^{s}\right){\left(X_{k}-{\hat{X}}_{k|k-1}^{s}\right)}^{T}\right]\right\}}^{-1}\;\left(\forall s\in M\right)$$
(48)$${\left\{E\left[\left(X_{k}-{\hat{X}}_{k,0}^{s}\right){\left(X_{k}-{\hat{X}}_{k,0}^{s}\right)}^{T}\right]\right\}}^{-1}\geq L_{k,0}^{s}={\left(P_{k,0}^{s}\right)}^{-1}$$
where ${\hat{X}}_{k,0}^{s}={\left(L_{k,0}^{s}\right)}^{-1}{\hat{Y}}_{k,0}^{s}$, $P_{k|k-1}^{s}={\left(L_{k|k-1}^{s}\right)}^{-1}$.The interaction covariance maintains consistency, that is
(49)$${\left\{E\left[\left(X_{k}-{\hat{X}}_{k,l+1}^{s}\right){\left(X_{k}-{\hat{X}}_{k,l+1}^{s}\right)}^{T}\right]\right\}}^{-1}\leq L_{k,l+1}^{s}\;\left(l=0,1,\cdots ,L-1\right)$$Hence, it can also be concluded that
(50)$${\left\{E\left[\left(X_{k}-{\hat{X}}_{k,L}^{s}\right){\left(X_{k}-{\hat{X}}_{k,L}^{s}\right)}^{T}\right]\right\}}^{-1}\geq L_{k,L}^{s}$$**Theorem** **2.**
*If the function ψ(·) is monotonically non-decreasing, and two positive semi-definite matrices $L_{1}$ and $L_{2}$ satisfy the condition $L_{1}\leq L_{2}$, then $0\leq \psi \left(L_{1}\right)\leq \psi \left(L_{2}\right)$ [24].*
From Theorem 2, it can be deduced that
(51)$$L_{k+1|k}^{s}=\psi \left(L_{k|k}^{s}\right)\leq \psi {\left\{E\left[\left(X_{k}-{\hat{X}}_{k|k}^{s}\right){\left(X_{k}-{\hat{X}}_{k|k}^{s}\right)}^{T}\right]\right\}}^{-1}={\left\{E\left[\left(X_{k+1}-{\hat{X}}_{k+1|k}^{s}\right){\left(X_{k+1}-{\hat{X}}_{k+1|k}^{s}\right)}^{T}\right]\right\}}^{-1}$$
Proof complete. □

## 4. Experiment and Result

To demonstrate the effectiveness of the DCIF, four experiments are conducted. The initial parameters of the bearing-only sensor network are shown in Table 1. The measurement errors are $\sigma_{\theta_{i}}=\sigma_{\varphi_{i}}=0.01\;\mathrm{rad}$. The target performs a uniform linear motion within 0∼25 s. Then the target performs a uniform acceleration linear motion within 25∼50 s, The acceleration of the target is $a_{x}$ = 10 m/s^2^, $a_{y}$ = 5 m/s^2^, $a_{z}=0$. At last, The target performs a uniform linear motion within 50∼100 s. Take the maneuvering frequency as μ=0.05, the maximum acceleration as $a_{\max}=10$ m/s^2^, the maximum acceleration probability as $P_{\max}=0.25$, and the probability that the acceleration is equal to zero as $P_{0}=0.75$. The sampling period T=1 s, it is assumed that the measurement periods of the nodes are the same, and the data are aligned in space and time [30].

In each simulation experiment, to mitigate the impact of initial parameters on the analysis results, the initial parameters of both the target and the bearing-only sensors are kept consistent [31]. To compare and analyze the effectiveness of the DCIF, the following simulation experiments are set up:

(1) A single bearing-only sensor is employed to track the target. The scenario is set to validate the observability conclusion analyzed in this paper.

(2) A pair of bearing-only sensors is employed to track the target. This scenario is indispensable regardless of the choice of state estimation method in the bearing-only sensor network.

(3) The proposed DCIF based on a bearing-only sensor network is employed to estimate the state of the maneuvering target.

To compare and analyze the superiority of the DCIF, the following simulation experiment is set up:

(4) The distributed unscented Kalman filtering (DUKF) method based on a bearing-only sensor network is employed to estimate the state of the target for comparison.

RMSE is commonly employed as a metric to measure filtering performance. Superior filtering performance minimizes the RMSE. The RMSEs of the spatial position and the motion parameters are utilized to assess the proximity of the estimation to the true state.

The RMSEs comparison of target position estimation are shown in Figure 3.

Within the set simulation time, the spatial position of the maneuvering target is estimated. The convergence speed and final error of the four methods are ranked from high to low as DCIF (1.38 × 10^3^ m) > DUKF (1.52 × 10^3^ m) > The estimation of the bearing-only sensor pair (1.75 × 10^3^ m) > The estimation of the single bearing-only sensor (2.92 × 10^3^ m). The method of using a single bearing-only sensor to estimate the target has a large error and cannot satisfy the expected results. The DCIF can better estimate the spatial position of the maneuvering target than other methods, which have the highest effectiveness and superiority.

The RMSEs comparison of target speed estimation are shown in Figure 4.

The convergence speed and final error of the four methods for estimating maneuvering target speed parameters are arranged from high to low as DCIF (4.62 × 10^1^ m/s) > DUKF (5.06 × 10^1^ m/s) > the estimation of the bearing-only sensor pair (61.5 × 10^1^ m/s) > the estimation of the single bearing-only sensor (7.68 × 10^1^ m/s).

The RMSEs comparison of target acceleration estimation are shown in Figure 5.

The convergence acceleration and final error of the four methods for estimating maneuvering target acceleration parameters are also arranged from high to low as DCIF (5.72 m/s^2^) > DUKF (6.15 m/s^2^) > the estimation of the bearing-only sensor pair (7.48 m/s^2^) > the estimation of the single bearing-only sensor (7.98 m/s^2^). The proposed DCIF method can better approximate the real acceleration of the maneuvering target as time goes on. More importantly, it can capture the sudden change of acceleration, which is significantly reflected in 10∼30 s and 40∼60 s, the system can faster respond to the strong maneuvering stage of target acceleration.

The results and analysis of the comparative simulation experiments are summarized as follows:Firstly, when using a single bearing-only sensor for state estimation, due to the unobservability of the system state, it is challenging to obtain the desired results. The simulation results of the first method can intuitively demonstrate that the system has difficulty converging within the simulation time.The results of observing a pair of bearing-only sensors that satisfy measurement constraints show a significant improvement, as seen in the simulation results of the second method. However, for maximum enhancement of observability and estimation accuracy, the bearing-only sensor network is a better solution. The comparison of state estimation methods in bearing-only sensor networks is evident in the third set with the DUKF method and the fourth set with the DCIF method.DUKF transfers nonlinear functions using the rules of unscented transformation, while DCIF employs the rules of cubature transformation. Unscented transformation is a nonlinear function transformation method based on sample selection. It selects a set of samples known as sigma points (SPs). Then, the nonlinear function is calculated on these SPs to generate new mean and covariance information through linear combination [32]. For some nonlinear functions, a large number of sigma points may be required to accurately approximate them, increasing computational complexity. In DCIF, the Gaussian integral is used to approximate the nonlinear function to obtain more accurate state estimation. Gaussian integration generates a set of CPs that are transformed by the covariance matrix to approximate the expectation value and covariance of the nonlinear function. DCIF does not require sample selection, and therefore, can better handle high-dimensional state spaces and highly nonlinear situations. DCIF generally performs better in bearing-only sensor networks and provides more accurate approximations of nonlinear functions.

## 5. Discussion

In the new mode of modern anti-stealth warfare, achieving effective detection is a prerequisite for gaining military advantages in conflict. Therefore, in the study of state estimation for stealthy maneuvering targets, designing superior and reliable filtering methods based on bearing-only sensor networks is an urgent and primary challenge. An early focus of our related work was on researching and experimenting with various filters but consistently failed to meet the expected estimation results. Subsequently, considering the impact of system uncertainty from the perspective of FIM, the novel DCIF method was introduced. This method deeply aligns with the observation constraints and distributed structure characteristics of the bearing-only sensor networks. Moreover, this paper not only rigorously proves the convergence consistency of the system through theoretical analysis, but also conducts comparative simulation experiments to validate the effectiveness and superiority of the novel DCIF. Considering that there are insufficient experimental conditions to support real comparative experiments, we have set up simulation experiments to indirectly validate our research results. In future work, we will further delve into the research on state estimation problems from two perspectives. On one hand, we will consider the impact of the target motion model in the Markov random jump system. On the other hand, we will address the time delay effects of bearing-only sensor networks in the filtering process.

## 6. Conclusions

This paper focuses on a novel sensor network model with two key characteristics: the distributed architecture and the bearing-only detection mode. The proposed DCIF tailored for this model is designed, combining CIF and the theory of coordinated consistency to estimate the state of spatial position and dynamic parameters. The CIF algorithm that satisfies the constraints of observation conditions acts on the local estimation of bearing-only sensor node pairs. Subsequently, DCIF is extended based on the weighted average consistency protocol to achieve global estimation for the bearing-only sensor network. The proposed DCIF method is rigorously theoretically proven, and its effectiveness and superiority for solving the target state estimation problem in bearing-only sensor networks are verified through comparative simulation experiments.

## Figures and Tables

**Figure 1 entropy-26-00236-f001:**
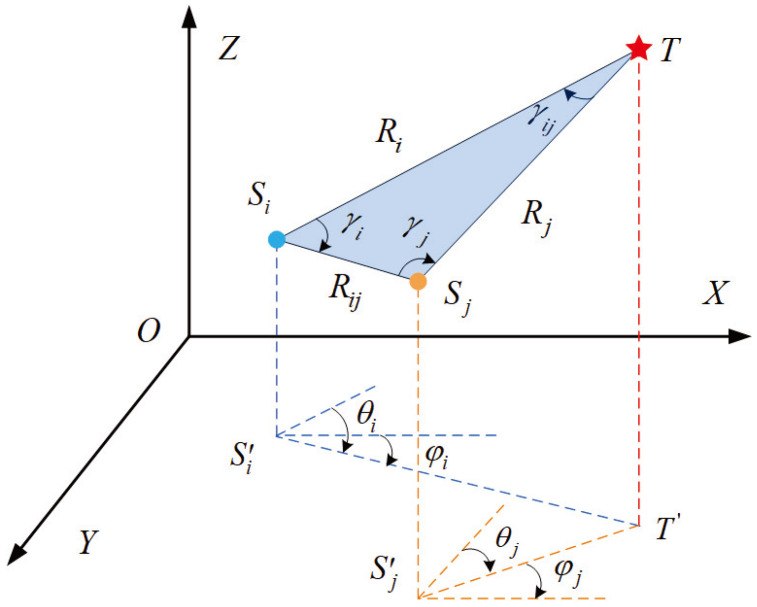
The system model.

**Figure 2 entropy-26-00236-f002:**
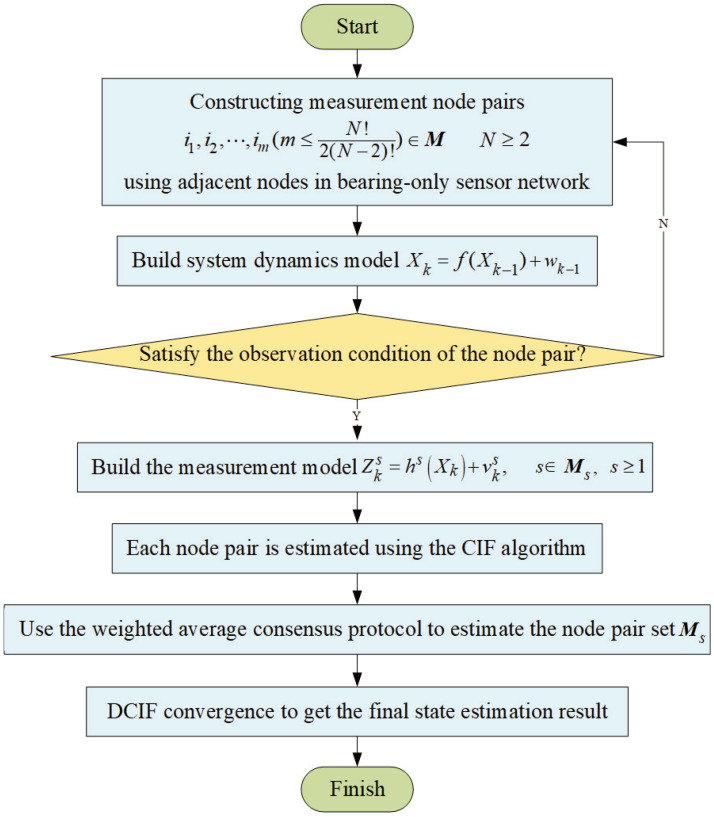
The flowchart of the DCIF method.

**Figure 3 entropy-26-00236-f003:**
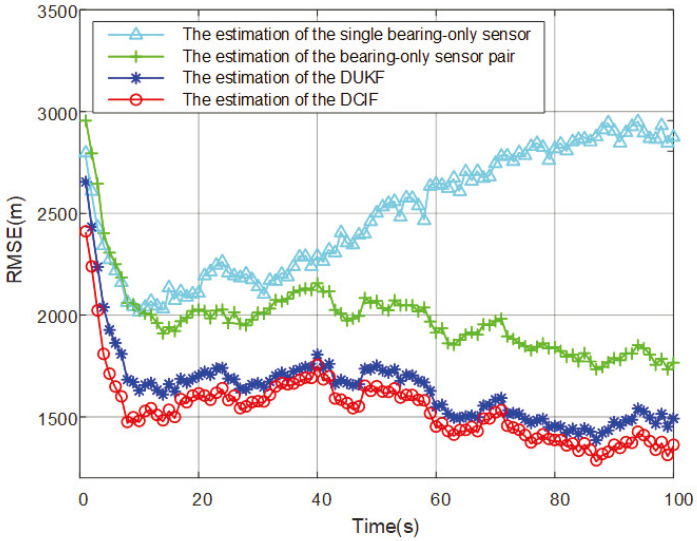
Comparison of RMSEs results of estimated spatial positions.

**Figure 4 entropy-26-00236-f004:**
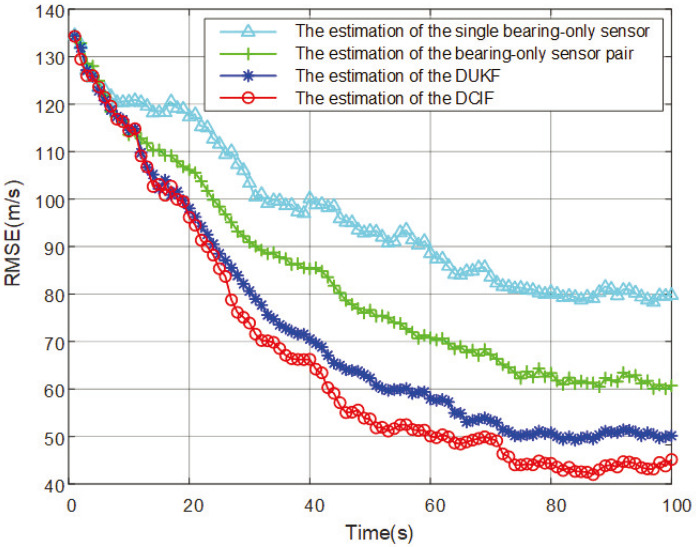
Comparison of RMSEs results for estimated speed parameters.

**Figure 5 entropy-26-00236-f005:**
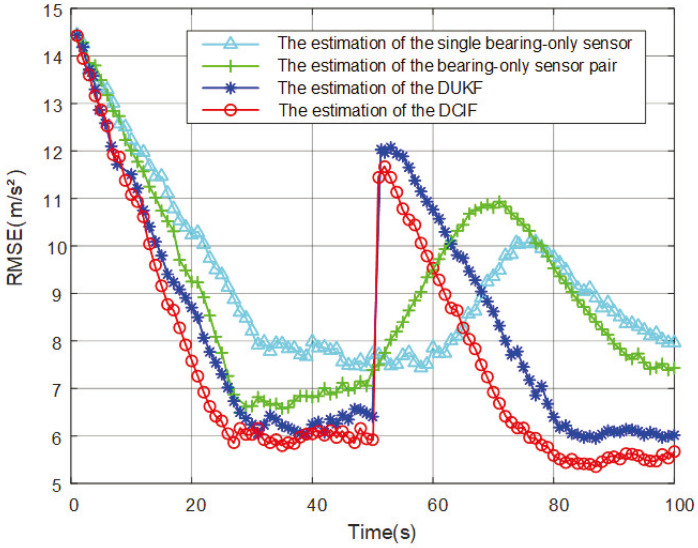
Comparison of RMSEs results for estimated acceleration parameters.

**Table 1 entropy-26-00236-t001:** Initial simulation experiment parameters.

	*x* (m)	*y* (m)	*z* (m)	*v_x_* (m/s)	*v_y_* (m/s)	*v_z_* (m/s)
Target	6.00 × 10^4^	4.00 × 10^4^	5.00 × 10^3^	2.50 × 10^2^	2.00 × 10^2^	1.00 × 10^1^
sensor1	2.50 × 10^3^	3.00 × 10^3^	4.00 × 10^3^	3.00 × 10^2^	2.50 × 10^2^	5.00 × 10^0^
sensor2	4.00 × 10^3^	7.00 × 10^3^	3.00 × 10^3^	2.50 × 10^2^	2.50 × 10^2^	5.00 × 10^0^
sensor3	1.00 × 10^4^	1.00 × 10^3^	1.00 × 10^3^	2.00 × 10^2^	3.00 × 10^2^	1.00 × 10^1^

## Data Availability

The data that support the findings of this study are available upon request from the authors.

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
