# Peer review of "Distributed Cubature Information Filtering Method for State Estimation in Bearing-Only Sensor Network"

_entropy, 2024, doi:10.3390/e26030236_

Round 1

Reviewer 1 Report

Comments and Suggestions for Authors

This paper deals with bearing-only tracking using filtering approach in the context of an array/network of sensors.

As the theoretical problem is easy to understand, the type of sensor that is considered is not clear. The application seems to be passive radar target detection so without any ranging information. Even if curvature KF is well known the novelty comes from the distributed approach. It would have been good to have a clear description of the 3 different presented approach : distributed / centralized / decentralized…

First section deals with the observation model. Equation (1) represent the target pose using two DOA measurements from 2 sensors. It seems that the detections are supposed to be synchronized. Is it realistic for a real application? With such assumption the problem become very simple and directly related to classical computer approach (bimonocular vision…). So from each pair of sensor can provide a complete 3D observation of the target except in case of bad parallax (no intersection of the DOA vectors).

The network of sensor is then decomposed as a graph and 2 by 2 neighbors are defined. Each pair of observation is then used in the filtering approach.

This process is very classical, two synchronized bearing observation from sensors with known locations are used to get 3D information.

Next section deals with a motion model discretized. Why did the authors has chosen to used the discret model and not the continuous one?

Section 3 focus on the DCIS filtering approach. Curvature filtering is first presented clearly then application to the current problem is proposed. More information on the « weighted average consistency protocol » is required here. How are multiple 2by2  observations fused? How to deal with outliers?

Proof of assumption 1 seems to be proof of assumption 4.

Result section :
- simulation is performed with synchonized and localized sensors with target evolving in a straight line.
- comparison is done with :
(i) a single bearing only sensor - why ???
(ii) two bearing only sensors - how are they chosen ?
(iii) the proposed algorithm
(IV) DUKF the same approch but with uncented KF instead of curvature.
It is very difficult to analyse the proposed results as the RMSE seems to be very big (>1,5 km) with 0.01 rad of error on the DOA observations. That seems huge but may depend on the size of the network and the distance to the target. What are these parameters? A full description of the simulation is needed.
Figure 5 is not analyzed. Where does the step in RMSE for acceleration comes? Are all the estimations consistent wrt the estimated covariance?
There is no comparison with standard EKF. It would have been more interesting than the single/dual BO examples.

Additional question : why are closest sensors of the network chosen as a pair instead of other configuration. In DOA estimation the most parallax you have the most accurate the estimation is.

The paper presents works on DOA estimation with multiple synchronized sensors. This estimation problem is not new, the novelty could come from the type of sensor of the type of sensor model but is not presented in the paper. Only a simulation with simple error of observed angles is used. The comparisons for the experiments are not pertinent and the two simple models should be replaced by other classical estimators.

Comments on the Quality of English Language

Some sentences don’t have verbs but it is easily understandable by a non native.

Reviewer 2 Report

Comments and Suggestions for Authors

The main contribution of this paper is to propose a distributed cubature information filtering (DCIF) method

Pag. 1, line 15: In the introduction section please clearly describe the goal of the paper and the novelty of it.

Pag. 7, line: In my opinion ‘white Gaussian with zero mean’ is not an appropriate choice! [1]

Pag. 10: I want a case study using realistic noise!

Pag. 13, section 4:

 In this paper, no experimental results are using real data. MDPI readers may want to see the results using real data rather than just synthetic data.

Conclusion

The contribution of the paper is not clear. The advantages over other algorithms are not presented and the experimental results are not compared to other approaches. So, the presentation of the results shall be improved to increase the impact of the document.

The contribution of this paper is weak!

References

[1] I. Shames, A. N. Bishop and B. D. O. Anderson, "Analysis of Noisy Bearing-Only Network Localization," in IEEE Transactions on Automatic Control, vol. 58, no. 1, pp. 247-252, Jan. 2013, doi: 10.1109/TAC.2012.2206693. keywords: {Equations;Noise measurement;Robot sensing systems;Noise;Jacobian matrices;Minimization;Vectors;Approximate localization;bearing;rigidity},

Round 2

Reviewer 2 Report

Comments and Suggestions for Authors

My reviews were taken into consideration, therefore I do not have more comments.